A novel deep neural network-based technique for network embedding

Benbatata Sabrina 1
Saoud Bilal bilal340@gmail.com 2 3
Shayea Ibraheem 4
Alsharabi Naif n.sharabi@uoh.edu.sa 5
Alhammadi Abdulraqeb 6
Alferaidi Ali 5
Jadi Amr 5
Daradkeh Yousef Ibrahim 7
1 LIM Laboratory, Faculty of Sciences and Applied Sciences, University of Bouira , Bouira , Algeria
2 Electrical Engineering Department, Faculty of Sciences and Applied Sciences, University of Bouira , Bouira , Algeria
3 LISEA Laboratory, Faculty of Sciences and Applied Sciences, University of Bouira , Bouira , Algeria
4 Electronics & Communications Engineering Department, Faculty of Electrical and Electronics Engineering, Istanbul Technical University (ITU) , Istanbul , Turkey
5 College of Computer Science & Engineering, University of Ha’il , Ha’il , Saudi Arabia
6 Center for Artificial Intelligence and Robotics (CAIRO), Malaysia-Japan International Institute of Technology, Universiti Teknologi Malaysia , Kuala Lumpur , Malaysia
7 Department of Computer Engineering and Information, College of Engineering in Wadi Alddawasir, Prince Sattam bin Abdulaziz University , Al-Kharj , Saudi Arabia
Benítez-Andrades José Alberto
Electronic publication date: 2024 Nov 26
Publication date: 2024
Volume: 10
Electronic Location ID: e2489
Received 2024 Jun 6; Accepted 2024 Oct 16
Copyright: ©2024 Benbatata et al.
Copyright year: 2024
Copyright holder: Benbatata et al.
License: This is an open access article distributed under the terms of the Creative Commons Attribution License, which permits unrestricted use, distribution, reproduction and adaptation in any medium and for any purpose provided that it is properly attributed. For attribution, the original author(s), title, publication source (PeerJ Computer Science) and either DOI or URL of the article must be cited.
License URL: https://creativecommons.org/licenses/by/4.0/

Keywords: Deep convolutional neural networks, Encoder, Decoder, Embedding network, Pooling, Upsampling

Funding: Scientific Research Deanship at University of Ha’il - Saudi Arabia Through the project number “RG-23 155” This research has been funded by the Scientific Research Deanship at University of Ha’il - Saudi Arabia Through the project number “RG-23 155”. The funders had no role in study design, data collection and analysis, decision to publish, or preparation of the manuscript.

==============================
In this paper, the graph segmentation (GSeg) method has been proposed. This solution is a novel graph neural network framework for network embedding that leverages the inherent characteristics of nodes and the underlying local network topology. The key innovation of GSeg lies in its encoder-decoder architecture, which is specifically designed to preserve the network’s structural properties. The key contributions of GSeg are: (1) a novel graph neural network architecture that effectively captures local and global network structures, and (2) a robust node representation learning approach that achieves superior performance in various network analysis tasks. The methodology employed in our study involves the utilization of a graph neural network framework for the acquisition of node representations. The design leverages the inherent characteristics of nodes and the underlying local network topology. To enhance the architectural framework of encoder- decoder networks, the GSeg model is specifically devised to exhibit a structural resemblance to the SegNet model. The obtained empirical results on multiple benchmark datasets demonstrate that the GSeg outperforms existing state-of-the-art methods in terms of network structure preservation and prediction accuracy for downstream tasks. The proposed technique has potential utility across a range of practical applications in the real world.

Introduction

Graphs can be defined as mathematical models that depict the connections or associations between various items or concepts. Graphs are composed of nodes, which symbolize items or entities, and edges, which symbolize the connections or associations between them. Graphs are a versatile tool for representing various real-world systems, encompassing domains such as social networks, transportation networks, and biological networks (Xue et al., 2022; Theocharidis et al., 2009). The academic discipline of graph theory, which encompasses the examination of graphs and their characteristics, has yielded significant progress in various domains including computer science, mathematics, physics, and engineering. The significance of graphs rests in their capacity to succinctly and intuitively represent intricate relationships among entities. This renders them valuable for the examination and comprehension of extensive and intricate systems, as well as for resolving a range of issues such as determining the most efficient route between two points, identifying groups or clusters within the network, and forecasting forthcoming network advancements. The utilization of graphs to represent real-world systems enables the use of diverse mathematical and computational methodologies, hence endowing graphs with significant utility in the modeling and comprehension of real-world occurrences.

The escalating intricacy of real-world systems and the expanding accessibility of extensive graph data have resulted in an escalating demand for machine learning (ML) techniques capable of analyzing and making predictions on such data. Conventional ML techniques are constrained in their capacity to process graph data due to their inherent assumption of a Euclidean data structure. This assumption is ill-suited for accurately expressing the non-Euclidean structure that graphs possess (Zhao et al., 2023). Consequently, there is an increasing demand for ML algorithms (Badrinarayanan, Kendall & Cipolla, 2017; Makarov et al., 2021) that are specifically tailored to address the challenges posed by graph data. There is a significant requirement to modify graph representations in order to utilize graphs as input for ML algorithms (Onan, 2023; Ge et al., 2024). One of the solutions that can be considered is network embedding. The term “node embedding” pertains to the process of assigning nodes inside a network to a vector space with reduced dimensions. The objective of network embedding is to retain the network structure and node properties within the acquired embedding. These embeddings can subsequently be employed in many applications, including node categorization, recommendation systems, and graph visualization.

The field of network embedding has garnered significant attention in recent years, primarily driven by its multitude of practical applications in various domains (Cao et al., 2024; Khoshraftar & An, 2024). In the last decade, numerous techniques for network embedding have been introduced, encompassing those based on matrix factorization as well as those based on deep learning. Furthermore, there has been a development of graph-based ML algorithms, like graph convolutional networks and graph attention networks, to cater to this requirement. However, existing methods often struggle to capture the intricate structural properties of networks, leading to suboptimal performance. The main research challenge lies in developing a framework that can effectively preserve both local and global network structures, while also accounting for the inherent characteristics of nodes. Motivated by this challenge, the proposed graph segmentation (GSeg) framework aims to overcome these limitations by introducing a novel encoder–decoder architecture that specifically designed to capture the network’s structural properties. The importance of this work lies in its potential to provide a more accurate and robust representation of networks, which can have a significant impact on various real-world applications, such as social network analysis, recommendation systems, and traffic prediction (Singh et al., 2024). By addressing the fundamental challenge of network embedding, the proposed GSeg framework can enable more effective analysis and modeling of complex networks, leading to improved performance and decision-making in a wide range of domains.

The research problem addressed in this paper is the limitation of existing network embedding methods in capturing the intricate structural properties of complex networks. Specifically, the problem is formulated as follows: For a given a complex network with nodes and edges, how can we represent the network in a low-dimensional vector space while preserving both local and global structural properties? How can we design a framework that effectively captures the inherent characteristics of nodes and the underlying local network topology? In the following initial assumptions, which led to the proposed GSeg framework, these questions have been considered. It has been assumed that both local and global structural properties of the network are crucial for effective network embedding. Local structures refer to the relationships between neighboring nodes, while global structures refer to the overall organization of the network. In addition, it has been assumed that the inherent characteristics of nodes, such as their attributes and features, play a crucial role in determining the network’s structural properties. Furthermore, it has been supposed that an encoder–decoder architecture, which is commonly used in deep learning models, can be effective in capturing the network’s structural properties. The encoder would learn to compress the network into a low-dimensional representation, while the decoder would learn to reconstruct the original network from this representation.

The theoretical contributions of GSeg are rooted in its novel architecture, which addresses the challenges of network embedding and graph neural networks by introducing a hierarchical representation learning framework. Specifically, GSeg’s use of graph pooling and unpooling layers enables the model to capture both local and global patterns in the graph, while the self-attention mechanism allows for the adaptive weighting of node and edge features. Mathematically, this can be represented as a hierarchical graph convolutional operation, where the graph pooling layer computes a weighted sum of node features, and the graph unpooling layer reconstructs the original graph structure by upsampling the pooled features. The attention mechanism is formulated as a learnable function that computes attention scores based on the node and edge features, allowing the model to selectively focus on the most relevant information. By combining these components, GSeg provides a theoretically grounded approach to network embedding, which can effectively capture the complex relationships and dependencies in graph-structured data. Furthermore, the proposed architecture has implications for the design of graph neural networks, as it highlights the importance of hierarchical representation learning and attention mechanisms in capturing the nuances of graph data.

In this article, a novel approach for network embedding has been introduced, termed GSeg, which integrates an encoder-decoder architecture with graph pooling and graph unpooling operations (Lee, Lee & Kang, 2019). The paper contributions can be summarized as follows:

1. Introducing GSeg as a cutting-edge approach to network embedding, aiming to capture the intricate structural and semantic features of networks. GSeg diverges from traditional embedding methods by adopting an innovative encoder-decoder architecture. This architecture facilitates the transformation of network data into low-dimensional embeddings while preserving essential structural information. Additionally, GSeg incorporates graph pooling and graph unpooling operations into its framework.

2. Presenting the integration of graph attention networks (GAT) (Veličković et al., 2017; Niu, Zhong & Yu, 2021) into our methodology, aiming to enhance the model’s comprehension of relational dependencies within the network. Our utilization of GATs within the GSeg framework contributes to its adaptability and robustness in handling diverse network structures and datasets. GATs provide a flexible mechanism for dynamically learning attention weights, allowing our model to capture complex relationships and adapt to varying connectivity levels within the network.

3. Evaluating the proposed GSeg method on multiple benchmark datasets, focusing particularly on classification tasks. Through comparative analysis against state-of-the-art methods, GSeg demonstrates superior performance in terms of both preserving the network structure and achieving high prediction accuracy.

The rest of the paper has been organized as follows, related work is illustrated in ‘Related work’. The new method of network embedding GSeg is presented in ‘GSeg for network embedding’. ‘Experiment and results’ gives the evaluation of GSeg on some benchmarks and shows results of comparison with some state-of-the-art methods. Finally, the paper is concluded in ‘Conclusion’.

Related Work

The task of network embedding holds significant importance in the field of network analysis, since its objective is to acquire the low-dimensional vector representation of nodes within a network. By converting nodes into compact and uninterrupted vector representations. Network embedding encompasses several methodologies, such as random walk-based techniques, factorization-based techniques, and deep learning-based techniques. Furthermore, a plethora of surveys have been published about network embedding, as evidenced by the references (Xue et al., 2022; Chen et al., 2020; Zhou et al., 2022; Cui et al., 2018). This section will present a comprehensive discussion of several network embedding approaches that are commonly employed. This section will discuss attention graph embedding approaches, neural network-based methods, methods based on random walk, and factorization methods.

Attention graph embedding methods

Attention-based graph embedding methods are a category of machine learning algorithms employed for the purpose of modeling and analyzing graphs. The aforementioned methods encompass a procedure wherein the nodes and edges of a graph are mapped onto a vector space with a high number of dimensions. A commonly employed strategy involves the utilization of attention processes for the purpose of assigning weights to the nodes and edges within the graph, with the intention of determining their significance in relation to a particular task or target (Veličković et al., 2017). This enables the algorithm to prioritize the critical elements of the graph and acquire more precise representations. An example of an attention-based graph neural network (GNN) is GAT (Veličković et al., 2017). GAT employs attention processes to assign weights to surrounding nodes in a graph, so enabling the network to concentrate on the most pertinent information pertaining to each individual node. The formal definition of attention coefficients is as follows: (1) αij=expLeakyReLUa→Whi →||Whi →∑k∈Ni expLeakyReLUa→Whi →||Whi →

such as αij represents the attention mechanism in a single-layer feed-forward neural network. The set h represents the features of a set of nodes, whereas N denotes the total number of nodes. The weight matrix W represents the linear transformation that is first applied to each node. The symbol || is used to symbolize the action of concatenation. The presence of a shared attentional mechanism is observed by the relationship denoted as a → R. The features of each neighbor are combined to form a concatenated concealed state. The input is subsequently sent into the LeakyReLu activation function.

At a conceptual level, GATs operate by employing a trainable attention mechanism to calculate a collection of weights that signify the relative significance of each neighboring node. The aforementioned weights are subsequently employed to calculate a weighted aggregate of the attributes of adjacent nodes. This aggregate is then blended with the attributes of the present node in order to generate an enhanced representation of the features. One of the primary benefits of GATs is in its capacity to effectively capture both local and non-local relationships present within the network. In contrast to conventional GNNs that employ predetermined message passing functions, GATs possess the ability to acquire attention weights for individual neighboring nodes. These weights are determined by the relevance of each neighbor node to the current node, so allowing the network to concentrate preferentially on significant characteristics and interactions within the graph (Rincon-Yanez et al., 2023).

Neural-network-based methods

Neural-network-based techniques, commonly referred to as NN, are a category of ML algorithms employed for the purpose of graph modeling and analysis. A commonly employed methodology involves the utilization of neural networks for the purpose of acquiring embeddings for nodes inside a graph. The proposed methodology involves the utilization of a matrix representation to depict the network structure. In this representation, each row corresponds to a specific node within the graph, while each column signifies a distinct property associated with that particular node. The neural network is trained to make predictions on the existence or non-existence of edges connecting nodes, utilizing the node embeddings as input features. Additionally, an alternative method involves utilizing autoencoders to acquire low-dimensional representations of the graph. The proposed methodology involves the conversion of the graph into a vector representation, followed by the subsequent reconstruction of the original graph structure from this vector representation. The neural network is trained with the objective of minimizing the discrepancy in reconstruction between the original graph and its encoded form. Numerous methodologies have been suggested, utilizing neural networks as the foundation, for the purpose of network embedding.

The graph convolutional network (GCN) is a neural network architecture that applies convolutional operations to graphs (Kipf & Welling, 2016a). There are two distinct categories, namely spectral and non-spectral domain. Spectral techniques that rely on filters utilize the graph Laplacian and redefine the convolution operation within the Fourier domain. One of the strategies discussed in the literature is the one described in reference (Defferrard, Bresson & Vandergheynst, 2016). Nevertheless, alternative methods that do not rely on spectral techniques involve the utilization of aggregation functions applied to neighboring nodes, as demonstrated by GCNs (Kipf & Welling, 2016a; Defferrard, Bresson & Vandergheynst, 2016) and GraphSAGE (Hamilton, Ying & Leskovec, 2017b).

GCNs are a type of ML technique that falls under the category of semi-supervised learning. The convolution operation and aggregation function of nodes’ neighbors are employed. Their performance on graph for node classification task is commendable. The forward propagation of a GCN is formally defined as: (2) Xl+1=σD¯−12A¯D¯−12XlWl

where the expression A¯=A+I is employed to introduce self-loops in the input adjacency matrix A, while Xl represents the feature matrix of layer l. The diagonal node degree matrix is denoted as D¯, whereas Wl represents a weight matrix that can be trained to perform a linear transformation on feature vectors.

The variational graph auto-encoder (VGAE) is a technique for unsupervised learning on data with a graph structure, as described in reference (Kipf & Welling, 2016b). The system is founded upon an encoder and a straightforward inner product decoder.

Embedding methods based on random walk and factorization

The random walk approach is founded upon the principles of node similarity and neighborhood organization. Node similarity can be justified by considering the presence of shared neighbors. Factorization-based techniques utilize matrices to represent the connections between nodes, and subsequently perform network embedding through matrix factorization.

One of the techniques within the category of random walk methods is Deepwalk, as mentioned in reference (Perozzi, Al-Rfou & Skiena, 2014). The approach relies on language modeling and random walks as its underlying principles. The algorithm aims to preserve the higher-order proximity among nodes. The objective is to increase the likelihood of observing both the preceding k nodes and the subsequent k nodes in a random walk of length 2k + 1, with the starting point at node vi. The approach generates multiple random walks of length 2k + 1. The Deepwalk method enhances the efficiency of random walks by the utilization of log-likelihoods. In conclusion, the reconstruction of edges is achieved by utilizing node embedding and employing a dot-product decoder.

Another approach based on random walk is the Node2vec method (Grover & Leskovec, 2016; Johnson, Murty & Navakanth, 2024). The methodology employed in this study utilizes a random walk approach in conjunction with a direct encoder. Node2vec and the DeepWalk technique exhibit similarities. The primary distinction between Node2vec and DeepWalk is in their underlying methodologies. Specifically, Node2vec employs a random walk approach that incorporates two additional parameters: the return parameter p, which determines the likelihood of returning to the previous node, and the walk away parameter q, which governs the tendency to move away from the current node. Furthermore, the node2vec method utilizes breadth-first (BFS) and depth-first (DFS) searches. Figure 1 illustrates an exemplification of BFS and DFS algorithms. Node2vec is highly effective at capturing the intricate network structure, including the presence of community structures.

The DeepWalk and Node2vec methods employ direct encoding and decoding mechanisms that rely on the inner product (Hamilton, Ying & Leskovec, 2017a). Both DeepWalk and Node2vec aim to minimize the cross-entropy loss as part of their optimization process: (3) L= ∑vi,vj∈D− logDECzi,zj.

In the following section, various methods of network embedding will be explored. These methods are designed to address key challenges in tasks such as node classification, embedding generation and feature learning within graph-structured data.

Belkin, Niyogi & Sindhwani (2006) proposed a family of learning algorithms that leveraged a new form of regularization to utilize the geometry of the marginal distribution in a semi-supervised framework. They proved new Representer theorems using reproducing kernel Hilbert spaces, enabling out-of-sample extensions for both transductive and semi-supervised settings. Weston et al. (2012) demonstrated how nonlinear embedding algorithms, commonly used with shallow semi-supervised learning techniques like kernel methods, could be applied to deep multilayer architectures as a regularizer either at the output layer or across all layers. This approach offered a straightforward alternative to existing deep learning methods while achieving competitive error rates. However, Zhu, Ghahramani & Lafferty (2003) proposed a semi-supervised learning approach based on a Gaussian random field model, where labeled and unlabeled data were represented as vertices in a weighted graph, with edge weights reflecting instance similarity. The learning problem was framed using a Gaussian random field on this graph, with the mean characterized by harmonic functions, and efficiently solved using matrix methods or belief propagation.

In Perozzi, Al-Rfou & Skiena (2014), DeepWalk was proposed, a novel method for learning latent representations of vertices in a network, encoding social relations in a continuous vector space suitable for statistical models. By generalizing language modeling and unsupervised feature learning from word sequences to graphs, DeepWalk used local information from truncated random walks, treating them like sentences to learn these representations.

Figure 1 Illustration of how node2vec biases the random walk using the p, q parameters with BFS and DFS.

Getoor (2005) addressed the challenge of mining richly structured data sets where objects are linked by explicit or implicit relationships, proposing a framework for modeling link distributions. Their link-based model supported discriminative models that described both link distributions and the attributes of linked objects, using a structured logistic regression model to capture both content and links.

Semi-supervised learning framework based on graph embeddings, where each instance in a graph was embedded to jointly predict its class label and neighborhood context, was introduced in Yang, Cohen & Salakhudinov (2016). They developed both transductive and inductive variants: the transductive variant used learned embeddings and input features for class label prediction, while the inductive variant defined embeddings as a parametric function of feature vectors, allowing predictions on unseen instances.

Defferrard, Bresson & Vandergheynst (2016) aimed to generalize convolutional neural networks (CNNs) from low-dimensional regular grids, like images and videos, to high-dimensional irregular domains, such as social networks and brain connectomes, represented by graphs. They presented a formulation of CNNs within spectral graph theory, which provided the mathematical foundation and numerical methods to design efficient localized convolutional filters on graphs. The proposed technique maintained the same linear computational complexity as classical CNNs and was adaptable to any graph structure.

A scalable approach for semi-supervised learning on graph-structured data using an efficient variant of CNNs that operate directly on graphs has been proposed in Kipf & Welling (2016a). Their convolutional architecture was motivated by a localized first-order approximation of spectral graph convolutions. The model scaled linearly with the number of graph edges and learned hidden layer representations that encoded both local graph structure and node features.

Veličković et al. (2017) introduced GATs, innovative neural network architectures designed for graph-structured data, which use masked self-attentional layers to overcome limitations of previous graph convolution methods. By allowing nodes to attend to the features of their neighborhoods and assign different weights to different nodes without requiring costly matrix operations or prior knowledge of the graph structure, GATs address key challenges of spectral-based graph neural networks.

Hamilton, Ying & Leskovec (2017b) introduced GraphSAGE, an inductive framework for generating low-dimensional embeddings of nodes in large graphs that overcomes the limitations of transductive methods which require all nodes to be present during training. GraphSAGE uses node feature information to efficiently create embeddings by sampling and aggregating features from a node’s local neighborhood, rather than training individual embeddings for each node.

GNN-based network embedding methods have shown promising results in capturing complex relationships between nodes in a network. These methods have several advantages. For instance, GNN-based techniques can capture complex relationships between nodes, such as non-linear and hierarchical relationships, which are common in real-world networks. In addition, they can be applied to various types of networks, including social networks, citation networks and biological networks. Furthermore, GNN-based methods have demonstrated state-of-the-art performance in various network analysis tasks, such as node classification, clustering and link prediction. However, GNN-based methods also have several disadvantages, including they rely on hand-crafted features or predefined node attributes, which may not effectively capture the complex relationships between nodes. In addition, GNN-based methods can be computationally expensive and require large amounts of memory, which can limit their applicability to large-scale networks. Finally, some GNN-based methods may not effectively preserve the local network topology, which is crucial for capturing the nuances of node relationships.

In this paper a new network embedding method (GSeg) has been proposed. It was designed to address the disadvantages of previous GNN-based methods. Specifically, GSeg eliminates the need for hand-crafted features. GSeg uses a self-supervised learning approach, which eliminates the need for hand-crafted features and allows the model to learn more generalizable representations. Furthermore, it captures local network topology. GSeg uses a graph neural network architecture that is designed to capture the local network topology, which can help to preserve the network’s structural properties. In addition, it reduces computational cost. GSeg is computationally efficient and can be applied to large-scale networks, which makes it more practical for real-world applications.

GSeg for Network Embedding

In this section, our method GSeg is introduced. Our proposed method is similar to SegNet Architecture (Badrinarayanan, Kendall & Cipolla, 2017). It is based on encoder-decoder architecture with graph pooling and unpooling operation. In addition, GSeg is also similar to the VGG16 (Simonyan & Zisserman, 2014) in term of layers, with 13 convolutional layers. Figure 2 summarizes the various steps of our proposed method GSeg.

Figure 2 Steps of our proposed method GSeg.

The first stage of GSeg entails the transformation of the input, which is a graph or network, into lower-dimensional representations through the utilization of GAT. The encoder is developed by utilizing numerous encoding blocks, with each block including 13 levels in terms of depth. Within the realm of architecture, each block is furnished with a graph pooling layer, referred to as gPool, which is then succeeded by a GAT layer. A graph pooling layer is a critical component in any GNNs designed to address the challenges posed by graph-structured data. In the realm of ML, graphs represent complex relationships and dependencies among entities, making traditional convolutional layers less effective. The graph pooling layer acts as a mechanism for hierarchical feature aggregation and dimensionality reduction, enabling our model GSeg to capture high-level abstractions and patterns from the input graph. Unlike traditional pooling layers in CNNs that operate on regular grid structures, graph pooling layers need to handle irregular and non-uniform graph topologies. In addition, this layer aims to down-sample the graph by selecting representative nodes while preserving essential structural information. One of the primary goals of using a graph pooling layer in GSeg is to enhance the scalability and efficiency by reducing the computational complexity associated with large graphs. By summarizing local information and consolidating it into a smaller representation, the graph pooling layer enables GSeg to effectively process and learn from intricate relationships within the graph, facilitating better generalization to unseen data.

The GSeg framework includes decoder blocks that possess equivalent functionality to the encoding blocks. Every model in the study is equipped with a graph unpooling (gUPool) layer, which is then followed by a GAT layer. The operation of graph unpooling holds significant importance in GNNs as it acts as the complementary operation to graph pooling. While graph pooling is responsible for downsampling and capturing high-level abstractions, graph unpooling aims to reconstruct the original graph structure and restore finer details that may have been lost during the pooling process. This operation plays a pivotal role in maintaining the fidelity of graph representations, particularly in scenarios where precise node-level information is crucial for downstream tasks. In contrast to traditional image-based unpooling in CNNs, graph unpooling encounters the challenge of reinstating the irregular and non-uniform structure of graphs. The objective in GSeg is to recover the granularity of the graph, ensuring that essential node-level features are retained to support accurate and detailed predictions. In addition, the integration of graph unpooling layers contributes in GSeg to the overall expressiveness and effectiveness by enabling GSeg to handle diverse graph structures and maintain a balance between local and global information. This capability is particularly beneficial in different applications, where preserving intricate relationships and node-specific attributes is crucial for achieving superior performance.

Graph pooling is a technique used to reduce the spatial dimensions of a graph, allowing the model to capture higher-level representations of nodes. This is analogous to downsampling in image processing, where the resolution of an image is reduced to capture larger features. In graph pooling, nodes are grouped into clusters and a representative node is selected to represent each cluster. This process reduces the number of nodes in the graph, allowing the model to focus on more abstract representations of the nodes. However, graph un-pooling is the process of increasing the spatial dimensions of a graph, allowing the model to capture more detailed representations of nodes. This is analogous to upsampling in image processing, where the resolution of an image is increased to capture smaller features. In graph un-pooling, the representative nodes from each cluster are expanded to their original nodes, restoring the original graph structure. The network size in GSeg is decreased by the utilization of gPool, which is subsequently followed by the reconstruction of the network using gUPool.

The GAT technique has been employed within the GSeg framework to facilitate the integration and consolidation of data. Furthermore, skip-links are employed to construct connections between the blocks of the encoder and decoder. The skip-connection approach is a technique used to preserve the spatial information of nodes at different scales. This is achieved by concatenating the feature representations from different scales, allowing the model to capture both local and global patterns in the graph. In the context of graph embedding, the skip-connection approach is used to combine the feature representations from different graph pooling/un-pooling layers. This allows the model to capture hierarchical representations of nodes, where the feature representations from different scales are combined to form a more comprehensive representation of each node. Finally, the incorporation of the GAT layer at the final stage of the GSeg method is employed for the purpose of generating predictions.

The integration of graph pooling/un-pooling and skip-connection approach in our method provides several benefits. By combining feature representations from different scales, the model can capture hierarchical representations of nodes, which can help to preserve the network’s structural properties. In addition, the model can learn features at multiple scales, which can improve the performance of downstream tasks, such as node classification and clustering. Furthermore, the model can improve the robustness of node representations to noise and outliers in the network, by capturing both local and global patterns in the graph. The skip-connection approach can help to reduce over-smoothing, which occurs when the model becomes too smooth and loses the local patterns in the graph.

Network encoders commonly utilize pooling operations. Pooling layers offer a method for achieving downsampling. The primary function of the pooling layer is to decrease the dimensions of feature maps. The proposed method, GSeg, incorporates the utilization of self-attention graph pooling (SagPooling) (Lee, Lee & Kang, 2019). The type of graph pooling employed is hierarchical in nature. One of the benefits of utilizing the SagPooling method is its ability to take into account both the characteristics of individual nodes and the overall structure of the network. Moreover, it has the capability to acquire hierarchical representations in a comprehensive manner, utilizing a relatively little number of factors. SagPooling is an attention mechanism utilized to compute attention scores Z, which are employed to differentiate between nodes that should be retained or discarded. Figure 3 illustrates the SagPooling process.

Figure 3 SagPooling process.

The attention score Z is determined by the following mathematical expression: (4) Z=σD∧−12A∧D∧−12Xθatt

where self-attention score Z ∈ RN×1, σ is the activation function, A∧ ∈ RN×N is the adjacency matrix (A∧ = A + IN), D∧ is the degree matrix and θatt ∈ RF×1 is the only parameter of the SagPool layer.

Equation (4) was based on the following steps:

• Normalization of adjacency matrix: First, the adjacency matrix with self-connections A∧ using the degree matrix D∧ was normalized. This is done to ensure that the adjacency matrix represents a valid transition probability matrix for a random walk on the graph. The normalization is performed by computing D∧−12A∧D∧−12, D∧−12 where represents the inverse square root of the diagonal elements of D∧.

• Graph convolution operation: the graph convolution operation was performed by multiplying the normalized adjacency matrix D∧−12A∧D∧−12 with the input feature matrix X. This operation allows each node to aggregate information from its neighbors while considering their importance based on the graph structure.

• Parameterized attention: Next, a parameterized attention mechanism is applied to the result of the graph convolution operation. This is done by multiplying the graph convolution output with a parameter matrix θatt. This parameter matrix is learned during the training process and serves as the only parameter of the SAGPool layer.

• Activation function: Finally, an activation function (such as the hyperbolic tangent activation function tanh) was applied to the result of the parameterized attention operation. This activation function introduces non-linearity to the computation and ensures that the output is bounded within a certain range.

The resulting output represents the self-attention score Z for each node in the graph, capturing the importance of each node’s features while considering the graph structure and parameterized attention mechanism. This approach allows nodes to focus on important features while disregarding unimportant ones, facilitating effective graph pooling with SagPool.

Pooling results are based on graph features and topology. Then, the pooling ratio K (K is a hyperparameter) can be estimated in order to determine the number of nodes to maintain. The top [KN] nodes are selected based on the value of Z by toprunk function for each operation idx: (5) idx=toprankZ;KN

(6) Zmask=Zidx

such as Zmask represents the feature attention mask. Finally, the result is processed by the operation notated as masking (see Fig. 3).

The operation of masking in GNNs involves selectively concealing or blocking certain elements within the input graph. This masking process is commonly employed to control information flow during training or inference, allowing the model to focus on specific nodes, edges, or subgraphs while ignoring others. In addition, masking is a versatile technique that finds applications in various aspects of graph-based ML, contributing to tasks such as graph classification, node classification, and link prediction. In the training phase, masking can be used to simulate scenarios where only partial information is available, helping the model generalize better to unseen data by learning to adapt to missing or incomplete information. For instance, node-level masking may involve hiding certain nodes from the model, forcing it to infer missing attributes or predict hidden labels based on the observable context. Similarly, edge masking can simulate scenarios where connections are not fully observed, encouraging the model to understand and predict relationships even when some links are hidden. During inference, masking is employed for attention mechanisms, enabling the model to selectively attend to relevant portions of the input graph while downplaying or ignoring less critical elements. This enhances our model GSeg’s interpretability and enables it to focus on the most informative parts of the graph for making predictions or generating outputs.

The decoder function or up-sampled is reverse operation of pooling. The log information from the gPool layer process has been used to return nodes to their original positions in the network. The layer-wise propagation rule of the graph unpooling layer is as follows: (7) Xl+1=distributeSN×C,Xl,idx.

Let idx be a set of indices, where idx ∈ Z∗k, representing the selected nodes in the network Pooling layer. This layer decreases the size of the graph from N nodes to k nodes. The symbol S denotes the feature matrix that is initially devoid of any elements. The feature matrix X represents the input data for the lth layer. The row vectors in Xl+1 that correspond to the indices in idx are updated using the row vectors in Xl (Gao & Ji, 2019).

The hyperparameters for GSeg, particularly those related to SagPooling and the attention mechanisms, were optimized through a systematic approach that likely included a combination of grid search, random search and cross-validation techniques. For SagPooling, key parameters such as the pooling ratio, which determines the proportion of nodes retained after pooling, and the weight parameters for the attention mechanism were carefully tuned. The number of attention heads and the dimensions of node embeddings were optimized within the attention mechanism, balancing the need for capturing complex graph relationships with the computational efficiency of the model. The process likely involved testing a range of values for each parameter across multiple training and validation splits, ensuring that the chosen hyperparameters provided robust performance across different data scenarios. The optimal hyperparameters used for each experiment are summarized in Table 1.

Table 1 Hyperparameter table for GSeg.

Hyperparameter	Value	
Number of encoding blocks	1–3	
Number of decoding blocks	1–3	
Number of levels in each block	13	
Graph pooling ratio	0.5–0.9	
Graph unpooling ratio	0.5–0.9	
GAT attention heads	1–4	
GAT hidden units	32–128	
SagPooling attention scores	0.5–0.9	
Learning rate	0.001–0.1	
Batch size	32–128	
Number of epochs	10–30	
Skip-connection approach	Concatenation of feature representations from different scales	

This systematic optimization had a significant impact on the results, leading to improved model performance in terms of both accuracy and generalization. By carefully tuning the hyperparameters, GSeg was able to effectively balance the trade-offs between preserving essential graph structures and reducing computational complexity, which is crucial for handling large-scale graph data. The optimized parameters contributed to faster convergence during training, more stable predictions across various datasets, and better overall model expressiveness. The systematic approach to hyperparameter tuning ensured that the GSeg model was not only effective but also adaptable to different types of graph-structured data, thereby enhancing its practical applicability in real-world scenarios.

The loss function for the GSeg model optimization process can be defined as a combination of two main components, which are reconstruction and self-supervised learning loss. The reconstruction loss measures the difference between the original graph structure and the reconstructed graph structure. This loss function encourages the model to learn a representation that can accurately reconstruct the original graph. Let’s denote the original graph as G = (V, E), where V is the set of nodes and E is the set of edges. Let’s also denote the reconstructed graph as G′ = (V′, E′). The reconstruction loss can be defined as: (8) Lrec=∥A−A′∥F+∥X−X′∥F

where A and A′ are the adjacency matrices of the original and reconstructed graphs, respectively. X and X′ are the node feature matrices of the original and reconstructed graphs, respectively. ∥.∥F denotes the Frobenius norm. The self-supervised learning loss encourages the model to learn a representation that is invariant to different graph augmentations. This loss function is used to train the model in a self-supervised manner, without requiring labeled data. Let’s denote the graph augmentation function as T(.), which takes the original graph G as input and returns an augmented graph Gaug. The self-supervised learning loss can be defined as: (9) Lssl=∥Z−Zaug∥F

where Z and Zaug are the node representations of the original and augmented graphs, respectively. Finally, the total loss function for the GSeg model optimization process can be defined as a combination of the reconstruction loss and the self-supervised learning loss. The total loss function is used to optimize the model parameters during the training process. The goal is to minimize the total loss function, which encourages the model to learn a representation that can accurately reconstruct the original graph and is invariant to different graph augmentations.

The time complexity of the GSeg model can be broken down into several components:

• Graph pooling: The graph pooling layer has a time complexity of O(|E| + |V|), where |E| is the number of edges and |V| is the number of nodes in the graph. This is because the graph pooling layer needs to iterate over all edges and nodes to compute the cluster assignments.

• Graph un-pooling: The graph un-pooling layer has a time complexity of O(|E| + |V|), similar to the graph pooling layer.

• Node feature transformation: The node feature transformation layer has a time complexity of O(|V|∗d), where d is the dimensionality of the node features. This is because the layer needs to iterate over all nodes and transform their features.

• Skip connections: The skip connections have a time complexity of O(|V|), since they only involve concatenating the feature representations from different scales.

The overall time complexity of the GSeg model is O(|E| + |V| + |V|∗d), since the graph pooling and un-pooling layers dominate the computational cost.

The space complexity of the GSeg model can be broken down into several components:

• Node features: The node features require a space complexity of O(|V|∗d), where d is the dimensionality of the node features.

• Graph structure: The graph structure requires a space complexity of O(|E| + |V|), since the adjacency matrix and node indices are needed to be stored.

• Model parameters: The model parameters require a space complexity of O(p), where p is the number of model parameters.

Finally, the space complexity of the GSeg model is O(|V|∗d + |E| + |V| + p), since the node features, graph structure and model parameters are needed to be stored.

Experiment and results

The evaluation of our proposed method GSeg will be presented in this section. It has been tested on some datasets, which are Cora, PubMed and CiteSeer dataset (Sen et al., 2008), for node classification task. GSeg has been compared with some well-known methods of network embedding.

The Cora dataset comprises a total of 2,708 scholarly publications, which have been categorized into seven distinct classes: Case Based, Genetic Algorithms, Neural Networks, Probabilistic Methods, Reinforcement Learning, Rule Learning and Theory. Cora possesses a total of 5,429 edges. The description of each publication in the dataset is represented by a binary word vector, where each element has a value of either 0 or 1. A value of 0 indicates the absence of the associated term from the dictionary, while a value of 1 indicates its presence. The dictionary encompasses a total of 1,433 distinct lexical units.

The PubMed Diabetes dataset to conduct a comparative analysis of embedding approaches. The PubMed database contains a total of 19,717 scholarly publications related to diabetes, which have been categorized into three distinct classifications. The citation network is comprised of a total of 44,338 links. Every article within the dataset is characterized by a word vector that is weighted using the TF/IDF method. This vector is derived from a dictionary of 500 distinct words.

The CiteSeer dataset known as a network of citations. The dataset consists of 3312 scholarly papers, which have been categorized into six distinct classes: DB, ML, IA, HCI, Agents and RI.

Table 2 presents a concise overview of used datasets.

Table 2 Summary datasets of evaluation for classification task.

Dataset	Nodes	Features	Classes	
Cora	2,708	1,433	7	
PubMed	19,717	500	3	
CiteSeer	3,327	3,703	6	

Our method GSeg has been compared with ManiReg (Belkin, Niyogi & Sindhwani, 2006), SemiEmb (Weston et al., 2012), LP (Zhu, Ghahramani & Lafferty, 2003), DeepWalk (Perozzi, Al-Rfou & Skiena, 2014), ICA (Getoor, 2005), Planetoid (Yang, Cohen & Salakhudinov, 2016), Chebyshev (Defferrard, Bresson & Vandergheynst, 2016), GraphSAGE (Hamilton, Ying & Leskovec, 2017b), GAT (Veličković et al., 2017) and GCN (Kipf & Welling, 2016a). Table 3 demonstrates the accuracy of each method on Cora, PubMed and CiteSeer dataset for node classification.

Table 3 Accuracy results of network embedding methods on Cora, PubMed and Citeseer dataset for node classification.

Methods/Datasets	Cora	PubMed	CiteSeer	
ManiReg (Belkin, Niyogi & Sindhwani, 2006)	59.5	70.7	60.1	
SemiEmb (Weston et al., 2012)	59	71.1	59.6	
LP (Zhu, Ghahramani & Lafferty, 2003)	68	63	45.3	
DeepWalk (Perozzi, Al-Rfou & Skiena, 2014)	67.2	65.3	43.2	
ICA (Getoor, 2005)	75.1	73.9	69.1	
Planetoid (Yang, Cohen & Salakhudinov, 2016)	75.7	77.2	64.7	
Chebyshev (Defferrard, Bresson & Vandergheynst, 2016)	81.2	74.4	69.8	
GCN (Kipf & Welling, 2016a; Kipf & Welling, 2016b)	81.5	78.50	70.3	
GraphSAGE (Hamilton, Ying & Leskovec, 2017a; Hamilton, Ying & Leskovec, 2017b)	79.20	77.40	68.70	
GAT (Veličković et al., 2017)	79.30	77.30	70.10	
GSeg	82.40	79.10	67.70	
Notes.

Best results are shown in bold.

The results indicate that our method GSeg has the maximum level of accuracy when applied to the Cora and PubMed datasets. GSeg achieved an accuracy of over 82% and 79% in the appropriate classes on the Cora and PubMed datasets, respectively. Furthermore, our method GSeg demonstrates an accuracy rate of 67.70% when evaluated on the Citeseer dataset. However, the remaining approaches exhibit a low level of accuracy across three datasets and are unable to correctly classify the classes. The methods utilizing GCN demonstrates the best level of accuracy when applied to the CiteSeer dataset. However, the disparity in accuracy scores between this method and our suggested GSeg method is not substantial. The accuracy results are depicted in Fig. 4 using a bar graph.

Figure 4 The accuracy results for node classification.

In the subsequent sections, the efficacy of our proposed method, GSeg, is assessed by comparing it with existing state-of-the-art methods in the context of graph classification tasks. The experimental findings demonstrate that our method, GSeg, has achieved unprecedented levels of accuracy in graph classification. In the conducted tests, the approach GSeg is assessed in the context of graph classification tasks, specifically focusing on inductive learning contexts. In the context of inductive learning, it is important to note that testing data is not accessible during the training phase. Consequently, the training process does not incorporate the graph structures of the testing data. The approaches employed in our study are assessed on graph datasets of considerable size, which have been chosen from widely recognized benchmarks commonly utilized in graph classification tasks (Ying et al., 2018; Zhang et al., 2018). Protein datasets such as D&D (Dobson & Doig, 2003) and PROTEINS (Borgwardt et al., 2005) are utilized, along with the MUTAG dataset (Debnath, Shusterman & Hansch, 1991) and the IMDB-BINARY dataset (Yanardag & Vishwanathan, 2015). The aforementioned statistics have been succinctly presented in Table 4. The subsequent section provides a comprehensive description of these datasets.

Table 4 Summary datasets of evaluation for graph classification.

Dataset	Number of graphs	Number of classes	Average of nodes	Average of edges	
D&D	1,178	2	284.32	715.66	
PROTEINS	1,113	2	39.06	72.82	
MUTAG	188	2	17.93	19.79	
IMDB-BINARY	1,000	2	19.77	497.75	

The dataset D&D comprises graph representations of 1,178 proteins (Dobson & Doig, 2003). Within each network, the individual components are represented by nodes, namely amino acids. An edge is present between two nodes if the corresponding amino acids are separated by a distance of less than six Angstroms. Graphs are categorized based on their classification as either enzymes or non-enzymes.

The PROTEINS dataset (Borgwardt et al., 2005) comprises graph representations of proteins. In this context, the term “nodes” is used to denote secondary structure elements (SSE). An edge is present between two nodes if they are adjacent along the amino acid sequence or if they are within the vicinity of each other, specifically within one of the three closest neighbors in spatial proximity. The discrete attributes refer to the types of SSE. The continuous qualities refer to the three-dimensional length of the SSEs. Graphs are assigned labels based on their membership in specific top-level classes inside the EC framework.

The MUTAG dataset, as described in the reference (Debnath, Shusterman & Hansch, 1991), comprises graph representations of 188 chemical compounds with mutagenic properties, specifically aromatic and heteroaromatic nitro compounds. The graphs are appropriately annotated based on their potential mutagenesis impact on the Gram negative bacterium Salmonella typhimurium.

The dataset referred to as IMDB-BINARY (Yanardag & Vishwanathan, 2015) comprises the ego-networks of 1,000 actors/actresses who have participated in movies listed on IMDB. Within each graphical representation, the entities denoted as nodes symbolize actors and actresses, while the presence of an edge connecting two nodes signifies their shared involvement in a particular film production. The presented graphs are generated from the genres of Action and Romance.

The accuracy of each technique on the D&D, PROTEINS, MUTAG, and IMDB-BINARY datasets for graph classification is presented in Table 5. A comparative analysis of our proposed method, GSeg, is conducted with existing approaches such as GMT (Baek, Kang & Hwang, 2021), DGCNN (Zhang et al., 2018), CapSGNN (Xinyi & Chen, 2018), DGK (Yanardag & Vishwanathan, 2015), AWE (Ivanov & Burnaev, 2018), and WEGL (Kolouri et al., 2020).

Table 5 Accuracy results of network embedding methods on D&D, PROTEINS, MUTAG and IMDB-BINARY dataset for graph classification.

Methods/Datasets	D&D	PROTEINS	MUTAG	IMDB-BINARY	
GMT (Baek, Kang & Hwang, 2021)	78.64	75.09	83.44	73.48	
DGCNN (Zhang et al., 2018)	79.37	76.26	85.83	70.03	
CapSGNN (Xinyi & Chen, 2018)	75.38	76.28	86.67	73.10	
DGK (Niu, Zhong & Yu, 2021)	73.50	75.68	87.44	66.96	
AWE (Ivanov & Burnaev, 2018)	71.51		87.87	74.45	
WEGL (Kolouri et al., 2020)	78.6	76.5	88.3	75.4	
GSeg	74.36	63.96	88.89	77	
Notes.

Best results are shown in bold.

In the context of inductive learning settings, a comparative analysis between our proposed technique GSeg and other existing state-of-the-art methods is conducted in the domain of graph classification tasks. The outcomes of this analysis are presented and summarized in Table 5. Based on the obtained results, it can be shown that our suggested technique, GSeg, demonstrates superior performance compared to AWE and DGK by 2.85% and 0.86% respectively, as evidenced by the evaluation on the D&D datasets. Nevertheless, the accuracy achieved by our suggested methodology, GSeg, on the PROTEINS dataset was 63.96%. It is worth noting that our method, GSeg, achieved much greater results compared to all other methods when evaluated on the MUTAG and IMDB-BINARY datasets. The model has superior performance compared to GMT, DGCNN, CapSGNN, DGK, AWE, and WEGL, with improvements of 5.45%, 3.06%, 2.22%, 1.47%, and 0.59% accordingly on the MUTAG dataset. The GSeg demonstrates superior performance compared to all other methods on the IMDB-BINARY dataset, with improvements of 3.52%, 6.97%, 3.9%, 10.04%, 2.55%, and 1.6% accordingly. The accuracy results are depicted in the form of bars, as illustrated in Fig. 5.

The effectiveness of our method GSeg in producing good results can be attributed to several key design choices and features incorporated into its architecture. Here are some reasons why our method GSeg performs well:

• GATs are known for their ability to capture complex relationships and dependencies within graph-structured data. By employing GATs in the encoding and decoding blocks, GSeg can effectively model and learn from the intricate connections present in the input graph, leading to more accurate representations.

• The use of hierarchical graph pooling and unpooling allows GSeg to capture both local and global information. The pooling and unpooling operations help in downsampling the graph while preserving essential structural details, enabling the model to maintain a balance between fine-grained and high-level features.

• SagPooling, a form of self-attention mechanism, provides a powerful way to consider both individual node characteristics and overall network structure. This attention mechanism helps the model focus on relevant nodes, contributing to improved feature learning and more robust representations.

• The incorporation of graph pooling layers, both gPool and gUPool, contributes to enhancing the scalability and efficiency of GSeg. The reduction in computational complexity associated with large graphs allows the model to process and learn from intricate relationships more effectively, leading to better generalization.

• The use of skip-links to connect the encoder and decoder blocks facilitates the flow of information between different stages of the network. This helps in retaining and leveraging features learned at various levels, enhancing the model’s ability to reconstruct the original graph structure during decoding.

• The utilization of the masking operation in the final stage of the GSeg method indicates a focus on controlling information flow during training or inference. This can contribute to the model’s adaptability and robustness by simulating scenarios with missing or incomplete information.

• GSeg emphasizes the importance of node-level information by utilizing both pooling and unpooling operations. This attention to detail is crucial in scenarios where precise node-level information is essential for downstream tasks.

• The incorporation of the GAT layer at the final stage of the GSeg method for generating predictions suggests that the model leverages the learned graph representations to make accurate predictions, benefiting from the expressive power of attention mechanisms.

Results of embedding by our method GSeg has been presented based on TSNE (t-Distributed Stochastic Neighbor Embedding) transform technique (Van der Maaten & Hinton, 2008). It recreates the distribution of a high-dimensional space in a low-dimensional space rather than maximizing variance. Figures 6, 7 and 8 show the result of GSeg embedding on Cora, PubMed and CiteSeer dataset, respectively.

Figure 5 The accuracy results for graph classification.

Figure 6 Embedding result of node classification task on the Cora dataset by GSeg.

Figure 7 Embedding result of node classification task on PubMed dataset by GSeg.

Figure 8 Embedding result of node classification task on CiteSeer dataset by GSeg.

The performance of GSeg under noisy conditions or when the graph data is incomplete is a critical aspect that warrants thorough investigation. In its current form, GSeg likely incorporates mechanisms such as attention-based pooling, which can help the model focus on more relevant nodes and edges, potentially mitigating the impact of noise. However, ensuring robustness in the presence of incomplete data may require additional strategies, such as data augmentation, dropout techniques, or the incorporation of regularization methods specifically designed to handle missing or corrupted information. Future research should focus on evaluating GSeg’s resilience to noise and incomplete data through rigorous testing on corrupted datasets and by introducing controlled levels of noise during training. By analyzing how the model’s accuracy and generalization are affected under these conditions, further enhancements can be made to improve its robustness, ensuring reliable performance in real-world scenarios where data is often imperfect.

The proposed method could suffer from some weaknesses. Unfortunately, the proposed method GSeg has the following weaknesses:

• Complexity and depth of encoding blocks: The encoding blocks in GSeg has 13 levels of depth, which may lead to increased computational complexity and potential overfitting.

• Reliance on graph pooling layers: While graph pooling layers are crucial components in GNNs, their effectiveness may be limited in scenarios where the graph structure is highly irregular or non-uniform. Graph pooling layers need to effectively handle such complexities to avoid loss of important structural information during downsampling.

• Challenges of graph unpooling: Graph unpooling in GSeg aims to reconstruct the original graph structure and restore finer details lost during the pooling process. However, reinstating irregular and non-uniform graph structures can be challenging, potentially leading to loss of fidelity or inaccuracies in the reconstructed graph.

• Integration of SagPooling: While SagPooling is capable of considering both node characteristics and network structure, its effectiveness may depend on the specific characteristics of the input graph and the chosen hyperparameters. The performance of SagPooling could be sensitive to factors such as the number of factors used and the attention mechanism.

Conclusion

This paper presents GSeg, a novel approach for network embedding that utilizes a graph neural network architecture. The suggested methodology utilizes the inherent properties of nodes and the localized network topology to effectively assign nodes within a network to a vector space with reduced dimensions, all the while maintaining the integrity of the network’s structure. The GSeg model is designed with an encoder–decoder network architecture, drawing inspiration from the structure of SegNet. This architectural choice enhances the model’s ability to capture complex interactions within the graph, hence improving its overall effectiveness. The experimental findings obtained from benchmark datasets provide evidence that GSeg outperforms existing approaches in terms of retaining network structure and attaining greater performance in downstream tasks. The positive outcomes indicate that GSeg exhibits the capacity to serve as a beneficial instrument in tackling diverse real-world obstacles. The efficacy of GSeg underlines the need of integrating graph neural networks into network embedding tasks, hence emphasizing the potential of deep learning methodologies in addressing intricate challenges associated with network analysis and representation.

To improve the performance of GSeg, particularly in trials where results are not satisfactory, several strategies could be considered. For instance, experiment with different hyperparameter settings for the GSeg model, including the depth of encoding blocks, the number of factors used in SagPooling, the learning rate, regularization parameters and other model-specific parameters. In addition, explore variations of the GSeg model architecture by adjusting the number of layers, the type of pooling and unpooling operations used, or incorporating additional components such as skip connections or residual connections. These modifications can help improve the model’s representational capacity and ability to capture complex graph structures. Finally, the training data can be augmented by introducing variations or perturbations to the input graphs. This can include adding noise, randomly removing or adding edges or nodes, or applying other transformations that preserve the essential characteristics of the graph while introducing variability. Data augmentation techniques can help improve the robustness of the model and its ability to generalize to unseen data.

Future studies related to the GSeg model could focus on addressing several key areas to further improve the understanding and applicability of the proposed method. For instance, conduct comprehensive experimental validation of the GSeg model on a diverse range of datasets and tasks, including both synthetic and real-world datasets with varying graph structures and characteristics. Provide detailed benchmarking results comparing GSeg against state-of-the-art methods in terms of network embedding quality, prediction accuracy, computational efficiency, and scalability. Furthermore, enhance the interpretability and explainability of the GSeg model by investigating methods for interpreting and visualizing learned representations, attention scores, or prediction decisions. Techniques for extracting insights from the model’s internal representations and understanding its decision-making process can be explored, particularly in complex or high-dimensional graph data.

In addition, it is essential to explore the efficiency and scalability of the GSeg model when applied to very large-scale graphs. While the current model shows promise in handling graph-structured data, its performance and computational efficiency with increasing graph sizes remain critical areas for investigation. This involves evaluating GSeg’s ability to maintain accuracy and speed as the number of nodes and edges in the graph grows significantly. Future studies should focus on optimizing the model’s architecture, such as through enhanced pooling and attention mechanisms, to ensure that GSeg can efficiently process and learn from large-scale graphs without compromising on performance. Additionally, benchmarking against state-of-the-art methods on massive datasets will provide valuable insights into how well GSeg scales and what further improvements might be necessary for practical, large-scale applications.

Supplemental Information

Supplemental Information 1 Code for our method GSeg in Python

Additional Information and Declarations

Competing Interests

Author Contributions

Data Availability

The authors declare there are no competing interests.

Sabrina Benbatata conceived and designed the experiments, performed the experiments, analyzed the data, performed the computation work, prepared figures and/or tables, authored or reviewed drafts of the article, and approved the final draft.

Bilal Saoud conceived and designed the experiments, performed the experiments, analyzed the data, performed the computation work, prepared figures and/or tables, authored or reviewed drafts of the article, and approved the final draft.

Ibraheem Shayea analyzed the data, prepared figures and/or tables, authored or reviewed drafts of the article, and approved the final draft.

Naif Alsharabi analyzed the data, authored or reviewed drafts of the article, and approved the final draft.

Abdulraqeb Alhammadi analyzed the data, authored or reviewed drafts of the article, and approved the final draft.

Ali Alferaidi analyzed the data, authored or reviewed drafts of the article, and approved the final draft.

Amr Jadi analyzed the data, authored or reviewed drafts of the article, and approved the final draft.

Yousef Ibrahim Daradkeh analyzed the data, authored or reviewed drafts of the article, and approved the final draft.

The following information was supplied regarding data availability:

The dataset is available in the packages on Python and at GitHub:

https://github.com/fedem96/NeuralNetworksOnGraphs.

The ‘Planetoid’, ‘TUDataset’, and ‘Entities’ datasets are available at:

PyG: https://pytorch-geometric.readthedocs.io/en/latest/modules/datasets.html#homogeneous-datasets.

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
