# Peer review of "A novel deep neural network-based technique for network embedding"

_PeerJ Computer Science, doi:10.7717/peerj-cs.2489_

## Round 0.1 · original submission · Major Revisions

Dear authors,
Thank you for submitting your manuscript to PeerJ. After a thorough review process, we have received feedback from the reviewers, and based on their detailed evaluations, we have decided to proceed with your submission contingent on major revisions.

Below is a summary of the key points highlighted by the reviewers:
1. Abstract and Introduction: Reviewer 2 noted that the abstract could better emphasize the main contributions of your work by removing redundant information. Additionally, the introduction should more clearly define the research challenges and the motivations behind your proposed methodology. This will help to frame your work within the context of existing challenges in the field.
2. Literature Review: Both reviewers suggested expanding the literature review to include discussions on recent advancements in Graph Neural Networks (GNNs) and network embedding techniques. Reviewer 2 specifically mentioned that while some preliminary background concepts can be omitted, a comparison of the pros and cons of recent techniques would strengthen your manuscript.
3. Methodology: Both reviewers raised concerns about the clarity and completeness of your methodology section. Reviewer 2 pointed out that several mathematical terms and notations require further explanation, and additional details should be provided on the integration of graph pooling/un-pooling and skip-connection approaches. Moreover, it is essential to discuss the time and space complexity of your model and clearly define the loss function used for optimization.
4. Experimental Design and Results: Reviewer 1 requested clarification on the optimization process for hyperparameters, particularly for SagPooling and attention mechanisms. Additionally, they recommended exploring how GSeg performs with very large-scale graphs and under noisy or incomplete data conditions. The inclusion of comparisons with other recent GNN advancements was also suggested.

We believe that addressing these points will significantly enhance the quality and impact of your manuscript. Once you have revised the manuscript, it will be sent back to the reviewers for a subsequent evaluation.

Please submit your revised manuscript through the PeerJ system within the suggested time-frame. If additional time is needed, do not hesitate to reach out.
We look forward to receiving your revised submission.

Reviewer 1 ·

Basic reporting

The paper introduces a new method called GSeg for network embedding. This method employs a graph neural network (GNN) framework to represent nodes in a network as vectors in a lower-dimensional space while preserving the network’s structure. The GSeg model uses an encoder-decoder architecture inspired by SegNet and incorporates graph attention networks (GAT), graph pooling, and unpooling operations.

Experimental design

The authors evaluate GSeg on several benchmark datasets, demonstrating its superior performance in terms of network structure preservation and prediction accuracy compared to state-of-the-art methods.

Validity of the findings

The authors evaluate GSeg on several benchmark datasets, demonstrating its superior performance in terms of network structure preservation and prediction accuracy compared to state-of-the-art methods.

Additional comments

1. How were the hyperparameters for GSeg, particularly for SagPooling and the attention mechanisms, optimized? Was there a systematic approach, and how did it impact the results?
2. Can the GSeg model efficiently handle very large-scale graphs, and how does its performance scale with increasing graph sizes?
3. The article mainly discusses network embedding. It is recommended to introduce related methods.
4. How does GSeg perform under noisy conditions or when the graph data is incomplete? Are there any specific measures taken to ensure robustness?
5. How does GSeg compare with other recent advancements in GNNs not covered in this paper, such as GraphSAGE or more recent versions of GATs?

Cite this review as

Reviewer 2 ·

Basic reporting

In this paper, authors proposed a novel technique calling as GSeg for dealing with networked data representation learning in which the graph neural network (GNN) has been utilized to capture the joined local proximity and underlying local network topological information. Thus, it can benefit the process of network node representation learning as well as fine-tuning for various graph-structured data mining tasks, such as classification. Specially, within this paper, authors have proposed an integration between various GNN-based architectures including the graph attention network, graph pooling/un-pooling layers and skip-connection to achieve the representations of network nodes in context of contrastive learning through auto-encoding architecture. By doing this, the proposed GSeg model is considered as sufficiently powerful to capture underlying network topological features through attention-driven graph propagation and convolutional operations. The GSeg is also presented as be capable for preserving complex relational dependencies within the information network to improve the task-driven adaptability and robustness of the given model while dealing with various network representation learning problems. Generally speaking, the proposed ideas in this paper is considered as interesting and unique in which various deep neural learning paradigms have been combined within the network embedding task and achieve richer-structural feature representations from the network to improve performance of various learning tasks. Extensive empirical studies within benchmark networked datasets have demonstrated the effectiveness of authors’ proposed ideas in this paper. Beside good points of this paper, I also have some revision suggestions as well as questions about the methodology of this paper, which might be useful for authors to improve their paper’s quality, including:
1) The contents within the abstract section should be revised in order to more highlight on the main/important contributions of authors’ proposed technique in this paper, some redundant contents related to the general approach/network embedding can be removed as they are quite common and not necessary to be presented in the abstract section.
2) For the contents of introduction section, more descriptions on the research challenges, motivations as well as how proposed ideas in this paper are considered as important to overcome these challenges – should be added. Additionally, authors should also add more explanations about the research problem formulation as well as initial assumptions that has led to their proposed ideas within this paper.
3) For the literature reviews within section 3, common preliminaries and background concepts related to GNN/graph attention, random walk on graph, etc. can be removed, as the readers can refer to these concepts within the original papers. In my opinions, authors should add more discussions about pros/cons of recent GNN-based network embedding techniques and which disadvantages of previous techniques are taken to be resolved in this paper.
4) For the methodology section, there are a lot of mathematical terms/notations are used without explanation – please check again and add missing descriptions for these terms/notations.
5) Please add more explanations about the integration of graph pooling/un-pooling and skip-connection approach, as this approach has been widely used within image segmentation model, e.g. U-Net, please explain more on how this approach can benefit the graph embedding process.
6) Moreover, the loss function for the model optimization process should also be provided within this section.
7) Finally, please also add the additional discussions about the time/space complexity of the given GSeg model within the last part of methodology section.

Experimental design

No comment.

Validity of the findings

Please refer to my basic reporting section.

Additional comments

No comment.

Cite this review as

---

## Round 0.2 · Minor Revisions

Dear authors,

Pay attention to the final suggestions from Reviewer 1

Best.

Reviewer 1 ·

Basic reporting

Pros:
1. The manuscript addresses an issue in the field of network embedding, which is a topic of considerable interest in the machine learning community. The focus on enhancing the accuracy and interpretability of network embeddings is timely and relevant.
2. The authors have conducted experiments on multiple benchmark datasets, which is a positive aspect of the research. The comparison with state-of-the-art methods provides a useful benchmark for evaluating the performance of the GSeg model.

Cons:
1. The manuscript claims to introduce a novel graph neural network framework named GSeg for network embedding. However, the proposed method appears to be a modification of existing techniques without a clear innovative aspect. The manuscript does not sufficiently differentiate GSeg from existing methods, and the theoretical contributions are not well-founded.
2. The methodology section lacks critical details regarding the implementation of the GSeg model. Key aspects such as the choice of hyperparameters, the rationale behind the number of intent subspaces, and the specific algorithms used for hypergraph construction are either not explained or are vaguely described.
3. As a research direction in machine learning and data mining on graphs, it is necessary to have some up-to-date surveys and related works

Experimental design

See above.

Validity of the findings

See above.

Additional comments

See above.

Cite this review as

Reviewer 2 ·

Basic reporting

I have carefully checked all authors’ revisions as well as feedback for my/other reviewers’ questions/recommendation and confirmed that all of problems in previous works have been sufficiently tackled. Thus, I thought this paper can be accepted for publication in this form. Thank.

Experimental design

No comment.

Validity of the findings

Please refer to my reviews for previous paper version.

Additional comments

No comment.

Cite this review as

---

## Round 0.3 · accepted · Accept

The authors have addressed all the comments.

Reviewer 1 ·

Basic reporting

The author addressed my concerns and I am inclined to accept.

Experimental design

The author addressed my concerns and I am inclined to accept.

Validity of the findings

The author addressed my concerns and I am inclined to accept.

Cite this review as